# Real-Life Experience of Continuously Infused Ceftolozane/Tazobactam in Patients with Bronchiectasis and Multidrug-Resistant *Pseudomonas aeruginosa* Infection in the Outpatient Setting

**DOI:** 10.3390/antibiotics12071214

**Published:** 2023-07-21

**Authors:** Francesco Venuti, Alberto Gaviraghi, Amedeo De Nicolò, Giacomo Stroffolini, Bianca Maria Longo, Alessia Di Vincenzo, Fabio Antonino Ranzani, Matilde Quaranta, Francesca Romano, Eleonora Catellani, Carlotta Marchiaro, Giacoma Cinnirella, Antonio D’Avolio, Stefano Bonora, Andrea Calcagno

**Affiliations:** 1Unit of Infectious Diseases, Department of Medical Sciences, University of Torino at the Amedeo di Savoia Hospital, ASL Città di Torino, Corso Svizzera 164, 10149 Torino, Italy; f.venuti@unito.it (F.V.); alberto.gaviraghi@unito.it (A.G.); biancamaria.longo@unito.it (B.M.L.); fabioantonino.ranzani@unito.it (F.A.R.); matilde.quaranta@unito.it (M.Q.); fr.romano@unito.it (F.R.); stefano.bonora@unito.it (S.B.); 2Laboratory of Clinical Pharmacology and Pharmacogenetics, Department of Medical Sciences, University of Turin, 10149 Turin, Italy; amedeo.denicolo@unito.it (A.D.N.); alessia.divincenzo@aslcittaditorino.it (A.D.V.); antonio.davolio@unito.it (A.D.); 3Department of Infectious-Tropical Diseases and Microbiology, IRCCS Sacro Cuore Don Calabria Hospital, Via Don A. Sempreboni, 5, 37024 Verona, Italy; giacomo.stroffolini@gmail.com; 4ASL Città di Torino, Amedeo di Savoia Hospital, Corso Svizzera 164, 10149 Torino, Italy; eleonora.catellani@aslcittaditorino.it (E.C.); carlotta.marchiaro@aslcittaditorino.it (C.M.); giacoma.cinnirella@aslcittaditorino.it (G.C.)

**Keywords:** ceftolozane/tazobactam, *Pseudomonas aeruginosa*, bronchiectasis, outpatient parenteral antimicrobial therapy, continuous infusion, therapeutic drug monitoring

## Abstract

(1) Background: Ceftolozane/tazobactam (C/T) is a novel β-lactam/β-lactamase inhibitor with excellent activity against the multidrug-resistant (MDR) *P. aeruginosa*. Continuous infusion (CI) dosing allows the optimization of pharmacokinetic and pharmacodynamic (PK/PD) properties of β-lactam antibiotics and may support patients’ treatment as outpatients. (2) Methods: Adult patients receiving their entire course of C/T as a CI in the outpatient setting were retrospectively included in the study. The primary outcome evaluated was clinical resolution. The secondary outcomes evaluated were PK/PD target attainment (ƒT > 4 × MIC) and microbiologic clearance at the end of treatment. Therapeutic drug monitoring to assess C/T concentration was performed. (3) Results: Three patients were enrolled in the study and received 9 g of C/T in CI every 24 h. One patient received an additional course of antimicrobial therapy due to disease exacerbation six months after initial treatment, accounting for four evaluated treatments. The primary outcome was achieved in 3/4 treatments and the secondary outcome was achieved in 4/4 and 3/3, respectively. In all patients, free ceftolozane concentrations were >10 times higher than the EUCAST breakpoint (4 mg/L). (4) Conclusions: Elastomeric infusion of C/T delivered in CI can be an effective and convenient way to treat acute diseases caused by MDR-*P. aeruginosa*, avoid hospital admission, and contribute to infection control strategies. Despite the small number of enrolled patients, clinical and microbiological results support this strategy.

## 1. Introduction

Ceftolozane/tazobactam (C/T) is a β-lactam/β-lactamase inhibitor antibiotic formed by a novel fifth-generation broad-spectrum cephalosporine with a well-known β-lactamase inhibitor [1], showing excellent activity against the multidrug-resistant (MDR) *P. aeruginosa* [2,3]. C/T has been approved in adult patients for the treatment of complicated intra-abdominal infections (cIAI); complicated urinary tract infections (cUTI), including pyelonephritis; and hospital-acquired bacterial pneumonia (HABP), including ventilator-associated bacterial pneumonia (VABP). Recommended dosing regimens in cUTI and cIAI are 1.5 g intravenously (IV) every 8 h and 3 g every 8 h in the case of HAPB/VAPB [4,5].

In healthy adult individuals, ceftolozane is almost exclusively excreted unmodified in urine. PK data showed a mean plasma half-life of 3 h, accounting for the need for multiple daily doses, while the volume of distribution of 13.5 L is close to the average extracellular volume, increasing the possibility of achieving therapeutic concentration levels in the extracellular compartment [1,5,6]. Tazobactam is also excreted through the kidneys, although metabolism to metabolite M1 has been observed. As for ceftolozane, the tazobactam volume of distribution is similar to that of the extracellular fluid volume (18.2), while the half-life is 1 h. Interestingly, the PK properties of tazobactam do not appear to be influenced by the coadministration of ceftolozane, as it was observed with piperacillin [5,6,7].

As for other β-lactams, C/T has a time-dependent bactericidal activity, which is optimal when the time (T) that the free drug concentration remains above the minimum inhibitory concentration (MIC) during dosing intervals (ƒT > MIC) is at least 40–70% of the total time of exposure [8,9]. However, more aggressive pharmacokinetic and pharmacodynamic (PK/PD) targets up to 100% ƒT > 4 − 5 × MIC may result in better outcomes [10,11,12]. Outpatient parenteral antimicrobial therapy (OPAT), through elastomeric infusion pumps, enables continuous infusion of antibiotics optimizing the PK/PD properties of β-lactams. C/T has a 24 h stability in aqueous solutions that allows for its OPAT implementation [13]. This administration route alongside PK/PD optimization favors cost-saving and antimicrobial stewardship interventions such as early hospital discharge and a reduction in hospitalizations and healthcare-associated complications [14].

Non-cystic fibrosis bronchiectasis is a chronic progressive respiratory disease characterized by irreversible bronchial airway dilatation and epithelial lining damage due to recurrent bacterial infections and continuous inflammation [15]. Clinically, the syndrome is characterized by sputum production, cough, dyspnea, and intermittent exacerbations that result in progressively decreasing lung function [16]. Exacerbations are associated with worse quality of life and increased socioeconomic costs due to frequent hospitalizations and subsequent mortality [17]. *Pseudomonas aeruginosa* is a non-fermenting Gram-negative aerobic bacterium that is frequently associated with chronic infection in bronchiectasis patients, and its presence represents a marker of disease severity [18,19]. Sputum cultures from bronchiectasis patients have been analyzed in different cohorts with samples resulting positive for *P. aeruginosa* in 15–30% of cases [20,21,22]. Bronchiectasis patients, especially if colonized by *P. aeruginosa*, receive frequent courses of antibiotics favoring the emergence of resistance to first-line antimicrobials recommended in bronchiectasis guidelines, such as fluoroquinolones [23,24].

The aim of this study is to evaluate the clinical and microbiological outcomes of outpatients with bronchiectasis treated with C/T through elastomeric pump infusion.

## 2. Results

A total of three patients received a continuous infusion of C/T during the study period and were included in our analysis. Patient characteristics, microbiology, and infusion regimens are summarized in Table 1. In the case of patient 1, two episodes were considered since they occurred during the study period and were separated by 6 months of wellbeing. Patient ages ranged from 65 to 75 years, and two of them were female. The median white blood cell count before treatment was 9105 cell/mL (range 7340–12,740 cell/mL), and the median C-reactive protein was 1.55 mg/dL (range 0.5–14.7 mg/dL and a cut-off value of >0.5 mg/dL).

All patients had MDR *P. aeruginosa* growth in a sputum sample (one *P. aeruginosa* mucoid strain was identified). Co-infections with other organisms were present in all patients with the exception of patient 3 and were selectively treated with concomitant antibiotic therapy. Co-infection with methicillin-susceptible *S. aureus* was found in patient 1 during both episodes and was treated with oral clindamycin during the first course of therapy. During the second episode, a course of inhaled amikacin was added, guided by the microbiological susceptibility of *P. aeruginosa*. Patient 2 had a history of multiple treatment failures for a *Mycobacterium avium* complex infection which was not treated since it was not considered the cause of the current exacerbation and was isolated in a single sample, as it did not play as much of a major role as in non-tuberculous mycobacterial pulmonary disease.

The *P. aeruginosa* MIC for ceftolozane/tazobactam ranged from 0.5 to 1 mg/L and was 0.125 mg/L in the case of the mucoid strain where a gradient diffusion method (Etest) was performed by our microbiology laboratory. The dosing regimens of 9 g of C/T diluted in 240 mL of normal saline infused over 24 h without a loading dose were consistently used in every treatment analyzed. To infuse the solution, medium long-term peripheral venous catheters (Midline) were inserted in patients 2 and 3, while for patient 1, a peripherally inserted central catheter (PICC) that had already been positioned for previous oncologic therapies was used.

### Primary and Secondary Outcomes

For the primary outcome, symptom resolution was obtained in three of the four treatments as documented by the OPAT clinical team at the end of treatment visit. Patient 1, at the end of the first treatment, was able to reduce oxygen supplementation to her baseline level and she did not report any adverse events related to C/T. During the second episode, patient 1 did not report treatment adverse events due to C/T but cough and headache after treatment with inhaled amikacin were described. Patient 2’s course of therapy was interrupted on day 11 since it was complicated by congestive heart failure, lobar pneumonia due to methicillin-resistant *S. aureus*, and thrombosis of the peripheral venous catheter (Midline), and she was hospitalized. These events were not considered directly related to C/T treatment, but early discontinuation was necessary, and a clinical resolution of symptoms was not obtained; thus, it was considered a treatment failure. Patient 3 had a clinical resolution of symptoms and no treatment adverse effects were reported but thrombosis of the peripheral catheter was clinically and radiologically confirmed. In all episodes that were considered clinically successful, C-reactive protein and white blood cells were in the range of normality at the end of treatment. While not considered directly related to C/T, thrombosis of the catheter was observed in two of the four treatments and required prolonged anti-thrombotic therapy after the end of treatment.

For the secondary outcome, microbiological resolution was obtained in three of the three treatments included in the analysis, since patient 2 was excluded from the intention-to-treat population for the microbiological outcome. Patient 1 could not produce sputa at the end of the first treatment while growth of polymicrobial flora was observed on the sputa obtained after the end of the second episode. The same type of growth was documented from the sputum sample of patient 3.

Pharmacokinetic (PK) analysis was performed for all patients on multiple days, and the timings and information are summarized in Table 2. All patients received 9 g of C/T in continuous infusion without a loading dose. The average ceftolozane AUC_24 h_ during the treatment ranged from 1348.1 to 2120.1 mg/L·h, with a median value among patients of 1418.3. The median average concentration of ceftolozane was 59.1 mg/L (IQR 56.2–88.3), with an intra-patient variability (percent coefficient of variation, CV%) of 8.1% (range 7.3–11.5) (Figure 1). Similarly, tazobactam AUC_24 h_ ranged from 174.1 to 293.4, with a median of 191.05 (Figure 2). The percentage of target attainment (ƒT > 4 × MIC for ceftolozane and ƒT > 0.5 mg/L for tazobactam) was 100% for both drugs.

Patients were monitored for recurrence during follow-up visits by the OPAT team and at the pneumology outpatient clinic. Microbiological follow-up consisted of one sputum sample collected approximately one month after the end of treatment. In the case of patient 1, the second encounter occurred six months after the end of the first treatment, but it was not considered as treatment failure due to the nature of bronchiectasis, a condition that predisposes to disease recurrence and exacerbations. Patient 2 was reassessed by our team after discharge and a sputum sample was obtained. Growth of *P. aeruginosa* was observed, with a lower microbial load (100,000 CFU/mL vs. 1,000,000 CFU/mL), but C/T susceptibility was not tested.

For patient 3, we repeated the culture on a sputum sample 17 days after the end of treatment, resulting in the growth of methicillin susceptible *S. aureus* and *Candida albicans*.

## 3. Discussion

Despite significant theoretical advantages, limited data have been published regarding the use of C/T in continuous infusion [25]. The first concern in this field is drug stability over time. The stability of C/T in aqueous solutions has been a major safety concern and it has been assessed in multiple studies. Raby et al. [23] compared the stability of C/T reconstituted in 0.9% sodium chloride with a volume of 240 mL (infusion rate 10 mL/h) in different dosing schedules: adjusted for renal function (0.45 g), standard (4.5 g), and HABP/VABP (9 g). Three different incubation temperatures were selected to simulate body temperature (37 °C), room temperature (25 °C), and refrigerated temperature (4 °C). Samples from all temperature conditions were taken at seven different time points within the first 48 h. For the refrigerated infusion, additional samples were taken up to 240 h. Ceftolozane and tazobactam stability remained > 90% at 24 h at every temperature, with the greatest loss at 37 °C. Refrigerated stability remained above 99% up to 7 days. However, the temperature of infusion devices depends on their position and proximity to the body and clothes and climate conditions; thus, some authors suggest using 32 °C for stability studies [26,27,28]. Terracciano et al. [29] assessed C/T stability when reconstituted and stored in elastomeric pumps with a volume of approximately 100 mL (AccuFlo and the I-Flow Homepump Eclipse) with a C/T recovery above 93% at 10 days.

In vivo clinical data on the use of C/T in continuous infusion as OPAT are still limited. Jones et al. [30] conducted a real-life study assessing C/T efficacy and feasibility as OPAT. Amounts of 4.5 g and 9 g of C/T were delivered using a Continuous Ambulatory Delivery Device in 240 mL 0.9% sodium chloride over 24 h. Symptom resolution was obtained in 85.7% of patients at the end of therapy. Microbiological resolution was demonstrated in three of seven patients. No patients reported adverse effects, and patient satisfaction was evaluated through a questionnaire revealing that all participants favored OPAT instead of hospitalization. A single-center retrospective analysis of the continuous infusion of C/T was conducted by Sheffield et al. [31] in seven adult patients with MDR *P. aeruginosa* deep-seated or device-related infections. In four patients, the continuous infusion of C/T was selected for transition of care in the outpatient setting with a dosing regimen of 6 g over 24 h, although the exact modality of administration in this setting was not specified. All participants had a resolution of symptoms, and in patients where TDM was performed, ceftolozane and tazobactam concentrations remained >4 × MIC and >0.5 g/mL, respectively, during the entire dosing interval. A recently published real-world multicenter study described outpatient experience with C/T in 126 patients with different types of infections [32]. In the cohort population, 18% of patients had respiratory tract infections; however, no patient with bronchiectasis exacerbation was reported. *P. aeruginosa* was the most frequent isolated pathogen accounting for 86.6% of the isolates, 45% of which were carbapenem resistant. Overall, the clinical success rate was 84.7% and adverse effects were reported in seven patients, including one discontinuation due to a non-specified catheter-related problem. C/T was administered in elastomeric pumps for self-administration at home or in polyvinyl chloride (PVC) for use with ambulatory or stationery infusion pumps at the office infusion centers. However, only in three patients (2%), C/T was administered in continuous infusion and TDM was not performed during treatment.

To our knowledge, our study is the first to assess the continuous infusion of C/T in the outpatients setting for the treatment of bronchiectasis exacerbation due to MDR-*P. aeruginosa* in real-life conditions. PK/PD analysis was conducted to evaluate the attainment of target concentrations in the clinical practice. Ceftolozane plasma concentrations remained abundantly above the target concentration (ƒT > 4 × MIC) for 100% of the time, showing a C_min_/4 × MIC over 2.5 in all cases, even considering the highest clinical breakpoint set by the European Committee of Antimicrobial Susceptibility Testing (EUCAST) and the Clinical and Laboratory Standards Institute (CLSI) for *P. aeruginosa* of ≤4 mg/L. Similarly, tazobactam concentration remained above the target >0.5 mg/L for 100% of the time. These results suggest that, in the context of OPAT, even a lower dose of C/T may be enough to reach sufficiently high drug exposure, with cost-saving benefits for the health system considering the high cost of the molecule.

Interestingly, the observed intra-individual variability in drug concentrations was quite low (<12% and <15% for ceftolozane and tazobactam, respectively), confirming the effectiveness of OPAT with the elastomeric pump to maintain a stable drug exposure throughout the treatment period.

The duration of treatment was decided according to available evidence and bronchiectasis guidelines [15,33], where a 14-day treatment course is recommended in the case of MDR *P. aeruginosa* exacerbation and when intravenous antibiotics are necessary. However, for the second treatment of patient 1, a shorter course regimen was selected according to the recent consensus [34] where shorter courses are recommended, especially in patients with less severe symptoms and rapid response to therapy. In our study, data on time to improvement were not collected, although patients were monitored on a daily basis by the OPAT team and clinical response was evaluated throughout.

The continuous infusion of C/T appears to be safe and well tolerated. However, our study highlights the potential risk of thrombosis in medium long-term peripheral catheters when positioned for an extended period of time for OPAT. This finding was not reported as an adverse event in any of the previous studies on the continuous infusion of C/T in the outpatient setting. Although we cannot generalize our experience, due to the limited sample size, we advise caution when using this type of catheter and strict monitoring of possible signs of thrombosis.

This study has a number of limitations such as the retrospective nature of the analysis and the small sample size. The concentrations were measured on different days for each individual, and the free concentration was only calculated correcting total concentrations using the theoretical protein binding. No PK comparison was possible with the standard infusion of C/T, so we cannot assess the actual increase in the overall exposure to the drugs obtained through continuous infusion. PK evaluation during the first day of infusion, before reaching the steady state, was lacking. However, bronchiectasis with exacerbation due to MDR-*P. aeruginosa* is not a frequent condition and our experience makes our analysis possibly interesting for future studies. Also, safety was not formally evaluated; thus, a careful approach is required when analyzing the results.

## 4. Materials and Methods

### 4.1. Participant Enrollment

This is a retrospective, single-center analysis of continuously infused C/T in the outpatient setting from January to November 2022. Adult patients (≥18 years old) with bronchiectasis pulmonary disease and sputum culture positive for MDR-*P. aeruginosa* (resistant to three or more classes of antimicrobials tested) were considered eligible for the study. The patient follow-up period for disease recurrence was at least 6 months after initiation and was conducted by the OPAT medical team in collaboration with the pneumology outpatient clinic. All patients received 9 g of C/T (6 g of ceftolozane and 3 g of tazobactam) every 24 h diluted in 240 mL of 0.9% sodium chloride at a rate of 10 mL/h using an elastomeric pump infusion device (BAXTER INFUSOR LV, 10 mL/h, ref 2C2036K). A 14-day treatment course was selected in case of the first episode of bronchiectasis exacerbation due to MDR *P. aeruginosa* while a 7-day course was chosen in case of a subsequent episode. Patients started treatment at the OPAT department of our hospital and presented at the facility during the following days, including weekends, for replacement of the device. The infusion was prepared by the pharmacy of our hospital every day before substitution. The preparation of the infuser was conducted following aseptic techniques under a sterile hood, in compliance with the norms of good preparation determined by the Italian “pharmacopoeia”.

Blood tests were performed at baseline and periodically during treatment, including complete blood count, C-reactive protein, transaminases, and serum creatinine. Microbiological details on susceptibility and culture were also obtained alongside therapy duration, dose, concomitant antibiotics, and adverse events.

The primary outcome was clinical resolution at the end of treatment, which was defined as a resolution of symptoms and return to the baseline status of bronchiectasis disease prior to the exacerbation. We assessed productive cough, dyspnea, and oxygen supplementation at the end of treatment visit, where related adverse events and early treatment discontinuation were also documented. Follow-up visits were organized in collaboration with the pneumology outpatient clinic to evaluate relapses.

### 4.2. Pharmacokinetic/Pharmacodynamic (PK/PD) Evaluation

The secondary endpoint was the evaluation of PK/PD target attainment and microbiologic clearance at the end of treatment. The PK/PD target for ceftolozane was described as the time during treatment, with free-drug concentrations exceeding four times the MIC (ƒT > 4 × MIC). Target attainment for tazobactam was defined as the duration of time during which the concentration exceeded 0.5 mg/L [35]. We used 21% (C) and 30% (T) as protein binding percentages in order to adjust the PK/PD calculation of ƒT > 4 × MIC [5,6,36]. To estimate the area under the concentration–time curve (AUC) in serum for ceftolozane and tazobactam, the linear trapezoidal rule was used via a non-compartmental analysis (NCA) using Phoenix WinNonlin software (Certara, Princeton, NJ, USA). To assess C/T serum concentrations, samples were collected on at least two separate days for each patient. In detail, blood samples were taken using heparinized tubes 15 min before substitution of the infusion device and 2 h after replacement on each day (Figure 1 and Figure 2). Plasma concentrations of ceftolozane and tazobactam were measured using a validated UHPLC-MS/MS method based on protein precipitation and internal standardization with stable isotope-linked ceftolozane, with a CE-IVDR marked kit (KIT-SYSTEM Antibiotics, CoQua Lab, Torino, Italy).

### 4.3. Microbiology

Microbiological clearance was defined as having one sputum sample collected at the end of treatment displaying no *P. aeruginosa* growth or not being able to produce a sputum sample. All *P. aeruginosa* isolates were identified using MALDI-TOF mass spectrometry and MICs were obtained using the Vitek 2 automated system antimicrobial susceptibility testing (AST)-N397 card (bioMerieux, Marcy-l’Etoile, France). One *P. aeruginosa* strain isolated from one of the patients expressed a mucoid phenotype, and susceptibility testing was subsequently performed with a gradient diffusion method (Etest).

## 5. Conclusions

The outpatient delivery of antimicrobials has been shown to be a safe [37] and cost-effective [38] measure to reduce hospital admissions, shorten hospital stay, improve patient satisfaction [39], and lower the barriers of access to care in disadvantaged patients in poverty-affected areas [40]. However, OPAT has a high level of complexity that requires planning by a multidisciplinary team that should include an ID specialist, a clinical pharmacist, a microbiologist, a clinical pharmacologist, and a nurse coordinator. To avoid readmission, clinical monitoring for treatment response, adverse events, and drug exposure is necessary during treatment.

C/T in continuous infusion using elastomeric devices has been demonstrated to be a safe and effective option for the treatment of bronchiectasis exacerbation due to MDR *P. aeruginosa*. TDM confirmed that this dosing regimen achieved target PK/PD concentrations, which is of great importance in the case of MDR Gram-negative bacteria and critical infections. As our study shows, TDM is an essential tool that should be always implemented where feasible to monitor β-lactam concentrations. Alongside pharmacological studies on the stability of C/T, this study can pave the way for the implementation of elastomeric infusion in the OPAT setting for multiple infections due to MDR-*P. aeruginosa*, with benefits for patients and a contribution to infection control strategies.

## Figures and Tables

**Figure 1 antibiotics-12-01214-f001:**
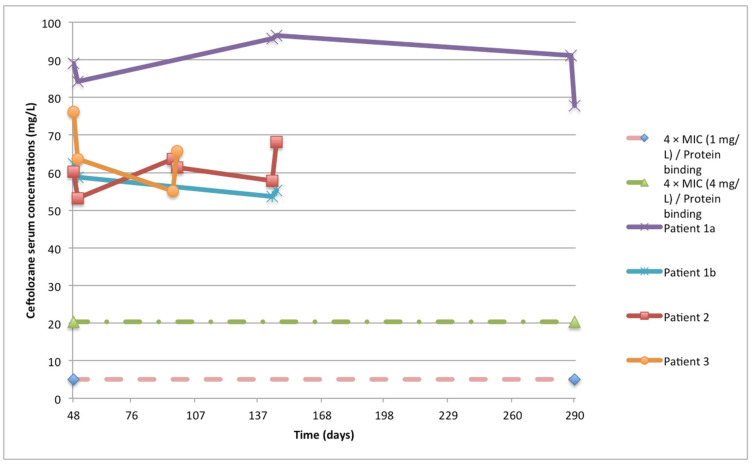
Total ceftolozane serum concentrations. The green line represents the target concentration calculated as 4 times the highest recorded susceptible minimum inhibitory concentration (MIC) of 1 μg/mL divided by the protein binding ratio for ceftolozane (0.79). The pink line represents the highest target drug concentration for any susceptible pathogen within the EUCAST breakpoint of ≤4 mg/L divided by the protein binding ratio for ceftolozane.

**Figure 2 antibiotics-12-01214-f002:**
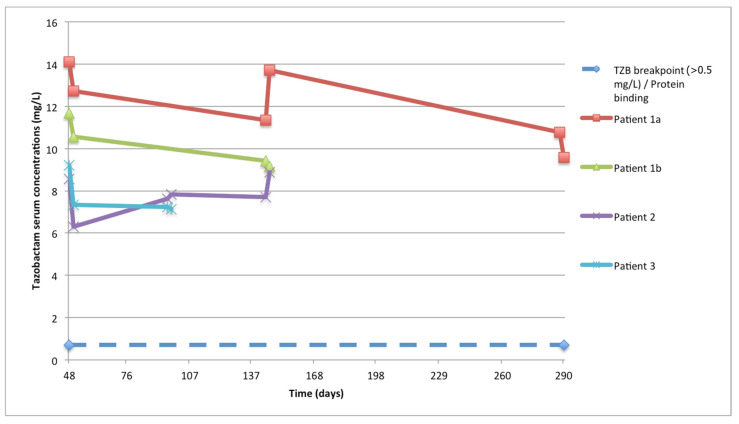
Total tazobactam serum concentrations. The blue line represents the target concentration for tazobactam >0.5 mg/L divided by the protein binding ratio for tazobactam (0.7). TZB, tazobactam.

**Table 1 antibiotics-12-01214-t001:** Clinical characteristics and outcomes of bronchiectasis patients receiving ceftolozane/tazobactam (C/T) in continuous infusion (CI) in the outpatient setting.

Patient ID	Age (Years)	*P. aeruginosa* MIC (mg/L)	Other Pathogens	Duration (Days)	Concomitant Antibiotics	Clinical Outcome	Microbiological Outcome (EOT)	AEs
Pt 1	75	1	MSSA	14	Clindamycin 600 mg q8h	Symptom resolution	Clearance	None
		1	MSSA	7	Aerosol amikacin 500 mg q12h	Symptom resolution	Clearance	Headache after inhaled amikacin
Pt 2	65	0.125	MAC	11	NA	Clinical failure	NA	Catheter thrombosis
Pt 3	69	0.5	Not present	14	NA	Symptom resolution	Clearance	Catheter thrombosis

MSSA, methicillin susceptible *S. aureus*; MIC, minimum inhibitory concentration; MAC, mycobacterium avium complex; ARs, adverse events; EOT, end of treatment; NA, not available.

**Table 2 antibiotics-12-01214-t002:** Summary of the PK and PK/PD evaluation of ceftolozane and tazobactam concentrations in plasma.

PK/PD Parameters for Ceftolozane												
Pt ID	C_max_ (mg/L)	C_min_ (mg/L)	Conc. Variability (CV%)	AUC_24h_ (mg/L·h)	AUC_8h_ (mg/L·h)	CL_ss_ (L/h)	MIC (mg/L)	Breakpoint MIC (mg/L)	ƒC_min_ (mg/L)	ƒC_min_/MIC (Observed)	ƒC_min_/4 × MIC (Observed)	ƒC_min_/4 × MIC (Breakpoint)
Pt 1	89.8	84.2	5.8%	2120.1	706.7	2.9	1	4	66.5	66.5	16.6	4.2
Pt 1 (2° treatment)	76.2	55.1	7.6%	1375.5	458.5	3.9	1	4	43.5	43.5	10.9	2.7
Pt 2	68.9	53.2	7.3%	1461.1	487.0	4.4	0.125	4	42.0	336.0	84.0	2.6
Pt 3	55.3	53.6	11.5%	1348.1	449.4	4.6	0.5	4	42.3	84.7	21.2	2.6
PK/PD Parameters for Tazobactam												
Pt ID	C_max_ (mg/L)	C_min_ (mg/L)	Conc. Variability (CV%)	AUC_24h_ (mg/L·h)	AUC_8h_ (mg/L·h)	CL_ss_ (L/h)	-	Target conc. (mg/L)	ƒC_min_ (mg/L)	-	ƒC_min_/C_target_	-
Pt 1	14.1	12.7	9.8%	293.4	97.8	9.6	-	0.5	8.89	-	17.8	-
Pt 1 (2° treatment)	9.2	7.2	9.7%	174.1	58.0	15.9	-	0.5	5.04	-	10.1	-
Pt 2	8.6	6.3	13.5%	174.8	58.3	17.2	-	0.5	4.41	-	8.8	-
Pt 3	9.4	8.5	11.2%	207.3	69.1	10.8	-	0.5	5.95	-	11.9	-

Pt, patient; PK, pharmacokinetic; PD pharmacodynamic, C_max_, peak serum concentration; C_min_, minimum serum concentration; AUC, area under the curve; CL_ss_, clearance at steady state; MIC, minimum inhibitory concentration; ƒC_min_, free minimum drug concentration; C_target_, target concentration.

## Data Availability

The data presented in this study are available on request from the corresponding author. The data are not publicly available due to privacy reasons.

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
