# Peer review of "Real-Life Experience of Continuously Infused Ceftolozane/Tazobactam in Patients with Bronchiectasis and Multidrug-Resistant Pseudomonas aeruginosa Infection in the Outpatient Setting"

_antibiotics, 2023, doi:10.3390/antibiotics12071214_

Round 1

Reviewer 1 Report

Venuti F. et al. is a report describing results from a retrospective study of continuously infused ceftolozone/tazobactam using elastomeric pumps in 3 patients with bronchiectasis and MDR-pseudomonas aeruginosa infection in the outpatient setting.

My comments are as follows:

1.      Page 3 Line 106: I suggest to provide a reference for the protein binding approximations for ceftolozone/tazobactam

2.      I would recommend adding information on pharmacokinetics of ceftolozone/tazobactam in the introduction section including their respective half-life

3.      I suggest reducing the discussion section to focus on how the findings from the current study advance the current knowledge of the subject and only briefly discuss relevant published literature.

4.      I suggest to clarify the duration of treatment in the methods section

5.      Figure 1: Why do the X and Y-axis value include (“) at the end? Suggest adding units for time on the X-axis label. Also the figure legend (both in the plot and the text below) has several typos and spelling mistakes which need to be fixed

Page 6 Line 201: For numbers described in the text use ‘.’ Instead of ‘,’ in all places to identify decimal digits

Please correct spelling mistakes and typos throughout the manuscript. Some examples below:

Page 7 line 223: ‘Bactericidal’ and ‘activity’ spellings need to be corrected.

Page 5 Line 163: Change form to from

Reviewer 2 Report

The authors should describe poperly which software were employed in order to model PK/PD data, as well, variations of models tested in order of evaluation can be done
